

# The spindle assembly checkpoint and speciation

Robert C. Jackson[1] and Hitesh B. Mistry[2]

[1] Pharmacometrics Ltd., Cambridge, United Kingdom
[2] Division of Pharmacy, University of Manchester, Manchester, United Kingdom

## ABSTRACT

A mechanism is proposed by which speciation may occur without the need to postulate geographical isolation of the diverging populations. Closely related species that occupy overlapping or adjacent ecological niches often have an almost identical genome but differ by chromosomal rearrangements that result in reproductive isolation. The mitotic spindle assembly checkpoint normally functions to prevent gametes with non-identical karyotypes from forming viable zygotes. Unless gametes from two individuals happen to undergo the same chromosomal rearrangement at the same place and time, a most improbable situation, there has been no satisfactory explanation of how such rearrangements can propagate. Consideration of the dynamics of the spindle assembly checkpoint suggest that chromosomal fission or fusion events may occur that allow formation of viable heterozygotes between the rearranged and parental karyotypes, albeit with decreased fertility. Evolutionary dynamics calculations suggest that if the resulting heterozygous organisms have a selective advantage in an adjoining or overlapping ecological niche from that of the parental strain, despite the reproductive disadvantage of the population carrying the altered karyotype, it may accumulate sufficiently that homozygotes begin to emerge. At this point the reproductive disadvantage of the rearranged karyotype disappears, and a single population has been replaced by two populations that are partially reproductively isolated. This definition of species as populations that differ from other, closely related, species by karyotypic changes is consistent with the classical definition of a species as a population that is capable of interbreeding to produce fertile progeny. Even modest degrees of reproductive impairment of heterozygotes between two related populations may lead to speciation by this mechanism, and geographical isolation is not necessary for the process.

## INTRODUCTION

The classical definition of a eukaryotic species is a population that is able to interbreed and produce fertile offspring. The ability to interbreed requires that sister chromatids of the two parents are able to align in the process of meiosis, and following recombination, segregate into daughter cells that each contain one and only one member of each pair of replicated chromosomes. Because the recombination process is not 100% faithful, misjoining occasionally occurs, with resulting chromosomal rearrangement. The rearranged

Corresponding author
Hitesh B. Mistry,
hitesh.mistry@manchester.ac.uk

chromatids are usually unable to align with their unchanged sister chromatids in a way that will produce gametes with a correct gene complement.

Since even closely related species usually differ in their karyotype, it has long been argued that chromosomal rearrangement may lead to speciation (*Mayr, 1970*; *Wilson, 2001*; *Bush, 1994*; *Todd, 1970*; *Dobzhansky, 1941*; *Ruffié, 1986*). The difficulty in this explanation of speciation is that when a chromosomal rearrangement arises in a single individual, if that individual mates with an unchanged partner their unmatched karyotypes will not produce viable zygotes. If enough individuals in a population generate the same chromosomal rearrangements in the same time and place, they could in principle form a new population, reproductively compatible with each other, though not with their ancestral population, and this would constitute a new species (*Parris, 2011*; *Parris, 2013*). However, the probability of this happening seems too low to explain observed rates of evolution.

The process of chromosome segregation during mitosis and meiosis is monitored by the spindle assembly checkpoint (SAC) (*Musacchio & Salmon, 2007*; *Sear & Howard, 2006*). This assures that daughter cells each receive one copy of each replicated chromosome. It has been argued that the SAC must be involved in speciation (*Kolnicki, 2000*; *Margulis & Sagan, 2002*), but at the time these papers were written SAC dynamics were not well understood, and the proposed mechanism did not answer the question of how the rearranged karyotypes were able to propagate. In recent years the SAC has been an object of intensive study, and its dynamics are increasingly well understood (*Sear & Howard, 2006*; *Kops, Weaver & Cleveland, 2005*; *Mistry et al., 2008*; *Mistry et al., 2010*; *Campbell & Desai, 2013*; *Sarangapani & Asbury, 2014*; *Britigan et al., 2014*; *Peplowska, Wallek & Storchova, 2014*; *Kim et al., 2013*; *Proudfoot et al., 2019*). The details of the SAC have been most extensively studied in mitosis. There are differences between mitosis and meiosis I (co-orientation) and meiosis II (bi-orientation) which have been reviewed by *Ohkura (2015)*. Our focus in the present discussion is on meiosis II, and we make the assumption that tension signalling is similar to that in mitosis.

Central to the operation of the SAC is the generation of a "wait" signal which suppresses progression to anaphase until all kinetochores are correctly attached by microtubules to the spindle poles. At this point, the wait signal rapidly decays, and the anaphase promoting complex/cyclosome (APC/C), released from inhibition, dissolves the cohesin links between sister chromatids, and the cell proceeds to anaphase. Several kinds of rearrangement are possible: in the following discussion we focus on fission, in which a single chromosome is broken in two, or fusion, in which two chromosomes that should be separate become joined together. A consideration of our present understanding of SAC dynamics shows how, in certain circumstances, chromosome fission or fusion can lead to speciation.

## Normal operation of the spindle assembly checkpoint

The SAC acts by sensing correct connections of kinetochores to the two opposite spindle poles. Until all kinetochores are attached, and in a state of tension, a "wait" signal is released that prevents progression to anaphase (Fig. 1). The wait signal is mediated by a complex group of microtubule-associated proteins, constituting the mitotic checkpoint complex (MCC) (*Musacchio & Salmon, 2007*). Of these, the most upstream is a bistable,

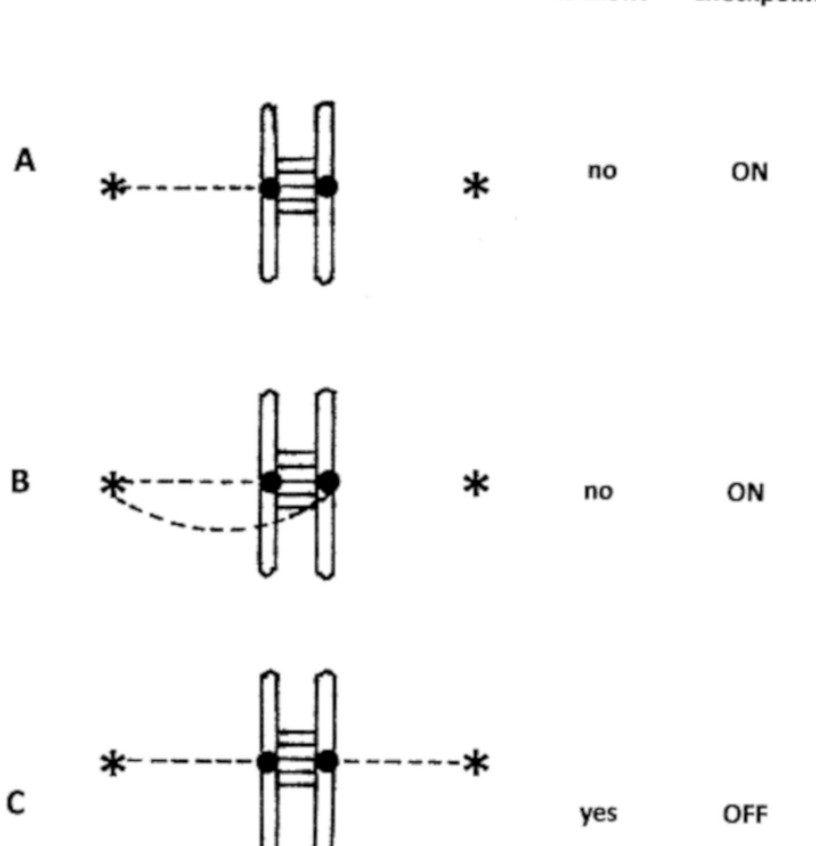

**Figure 1  Normal operation of the SAC.** (A) Monotelic; (B) syntelic; (C) amphitelic. The vertical bars represent sister chromatids of germ cells in the second meiotic division, attached to each other by cohesin. The filled circles represent kinetochores, and the asterisks are spindle poles. Dashed lines indicate micro-tubules.

tension-sensitive enzyme, believed in insects to be aurora, or in vertebrates the homologous aurora kinase B (*Campbell & Desai, 2013*). Either over-expression of aurora kinase B, or its inhibition in mammalian cells results in aneuploidy (*Gonzalez-Loyola et al., 2015*; *Wang et al., 2010*). Microtubules grow from the opposite spindle poles, and attach randomly to the kinetochores (in fact each kinetochore has 20 –30 microtubule binding sites). When the kinetochore of one of a pair of sister chromatids is attached to a single spindle pole, the situation described as "monotelic" in Fig. 1, the kinetochore is not in a state of tension. Now when a microtubule attaches to the other sister chromatid, we assume that there is a 50% chance that it will be attached to the opposite spindle pole (amphitelic attachment), and a 50% chance that it will attach to the same spindle pole as its sister (syntelic attachment). Syntelic attachments are not in a state of tension, which results in the second attachment being removed through activity of the aurora enzyme (*Sear & Howard, 2006*; *Kops, Weaver & Cleveland, 2005*; *Mistry et al., 2008*; *Mistry et al., 2010*). After this, there is again a 50:50 chance that the second kinetochore will become correctly attached. Amphitelic attachments

are in a state of tension, aurora kinase B is inactive, so that a particular sister chromatid pair does not produce a wait signal. When the last pair of sister chromatids in the cell is correctly attached, the wait signal rapidly decays, and the cell progresses to anaphase, in which the cohesin bonds between chromatids dissolve, and the cell moves to telophase.

In addition to monotelic, syntelic, and amphitelic attachments, a fourth state is possible, merotelic attachment, in which one kinetochore is correctly attached while the other is simultaneously bound to microtubules from both spindle poles. Merotelic attachments do not appear to be detected by the SAC, and are likely to lead to abnormal mitoses or meioses. For the purpose of our analysis, we assume that the frequency of merotelic attachments is the same for cells with normal or rearranged karyotypes, in which case they would not influence the relative incidence of zygotes with a normal gene complement.

## Operation of the SAC following chromosomal fission or fusion

Now consider the situation discussed by *Kolnicki (2000)*, in which kinetochore duplication has resulted in a particular chromosome being split into a pair of smaller, acrocentric chromosomes. Figure 2 describes the events in metaphase II of meiosis in this cell. Any syntelic attachments formed prior to the attachment of the second acrocentric will be reversed. At the stage where the single un-fissioned chromatid and one of the acrocentrics have become amphitelically attached, two possible configurations (or their mirror images) will result. These are shown as Figs. 2A and 2B. For each of these configurations, there is a 50% chance that the remaining unattached kinetochore will become correctly attached, resulting in configuration C. Of the remaining 50%, 25% will attach to give configuration D, and 25% will give configuration E. Systems C, D, and E are all in tension, and will progress to anaphase. Configuration C will yield equal numbers of gametes with the karyotypes shown as type I (Fig. 3A) and type II (Fig. 3B). Configuration D will yield equal numbers of the gametes labelled type III and type IV. Configuration E will result in equal numbers of gametes type V and type VI (Fig. 3). Gametes of type III and type V are missing many genes, and will probably be nonviable. Gametes of type IV and type VI contain a complete complement of genes, and may progress to fertilisation.

When fertilisation is by wild-type gametes, the possible resulting zygotes are as illustrated in Fig. 4. Fertilisation of gametes of type I gives wild-type zygotes (Fig. 4A). Fertilisation of type II gametes gives zygotes that are heterozygous for the chromosomal fission (Fig. 4B). These contain a full gene complement, and normal gene dosage. These zygotes are expected to be fully viable, and their fate will be considered below. Gametes of types IV and VI, after fertilisation by a wild-type gamete, will give zygotes of types C and D). These have abnormal gene dosage, and are likely either to spontaneously abort, or to develop into offspring with birth defects. These surviving individuals will not necessarily be sterile, but are likely to have at least partially decreased fertility.

In summary, when neokinetochore formation, and subsequent chromosomal fission, occur in meiosis 25% of the resulting gametes will receive the normal monocentric chromatid. Of the remaining 75%, one-third result in gametes with a pair of acrocentric chromosomes. These cells will be stable, and segregate correctly at mitosis. The other 50% will give half (25% overall) gametes with missing genes and half (25%) gametes

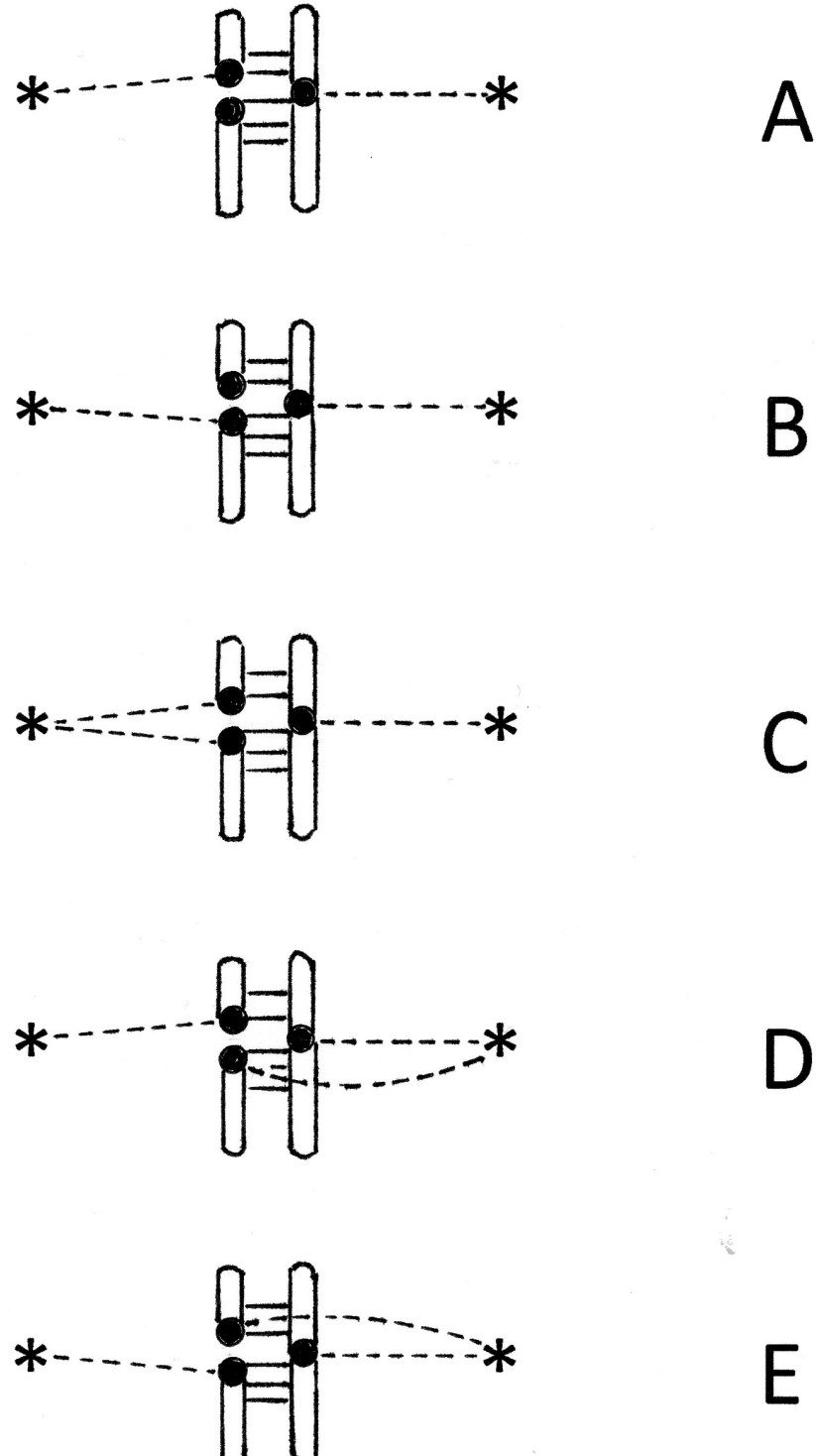

**Figure 2  Operation of the SAC following a chromosome fission.** Symbols are defined in the legend to Fig. 1. (A–E) represent the possible attachment modes that are under tension and will lead to gamete formation as described in the text.

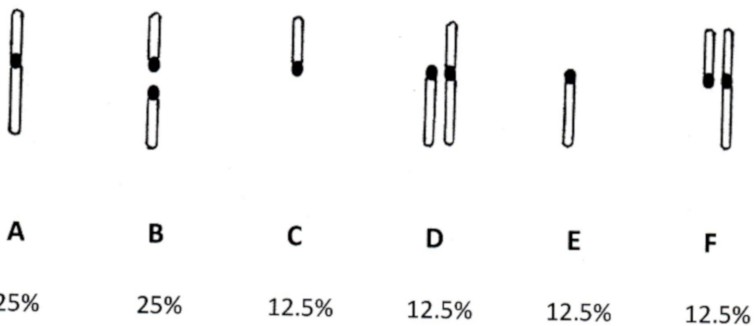

**Figure 3  Fission karotypes.** Karyotypes of gametes formed from meiosis of a cell in which karyotypic fission has occurred. The lower row of figures indicates the fraction of total gametes with the indicated karyotype. (A) type I; (B) type II; (C) type III; (D) type IV; (E) type V; (F) type VI.

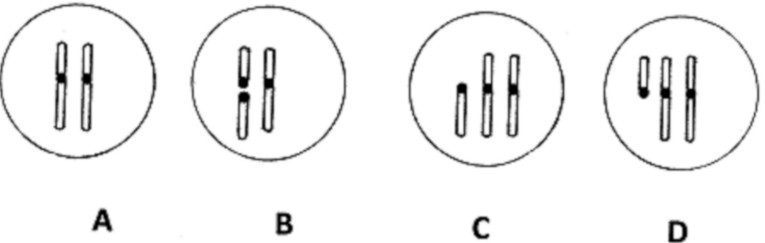

**Figure 4  Creation of zygotes.** Karyotypes of zygotes resulting from fertilisation of gametes of types I, II, IV and VI. (A) Two normal chromosomes; (B) one normal and two small acrocentric chromosomes; (C and D) Two normal and one small acrocentric chromosomes.

with incorrect gene dosage. If neokinetochore formation occurs without chromosomal fission, the result will be 50% of gametes with a bicentric chromosome. For the reasons discussed by *Kolnicki (2000)*, these chromosomes will have a 50% chance of splitting into two acrocentrics at each subsequent mitosis. The overall effect on fertility will be the same as the situation discussed above, in which chromosomal fission immediately follows neokinetochore formation.

A similar analysis may be performed for the situation where two chromosomes fuse. If the fused chromosome is capable of aligning during meiosis with the two corresponding wild-type chromosomes, then viable and non-viable gametes will result in the ratios discussed for chromosomal fission. Obviously if the fused chromosome is incapable of alignment with its cognate parental chromosomes, the chromosomal rearrangement will be lethal.

## METHODS

The *speciation algorithm* is a form of genetic algorithm (*Holland, 1975*; *Holland, 1998*) (Fig. S1). A growth equation calculates reproduction rates for the various karyotypes. The objective function determines how many individuals of each karyotype survive to

reproductive age in the next generation, and also how many individuals migrate between ecological niches. Positive feedback from the objective function to the growth equation indicates that those individuals best adapted to their environment represent a higher proportion of the next generation. Information on fertility of karyotypic variants is reflected in growth rates used by the growth equation; for heterozygotes, fertility rates following chromosomal re-arrangements (including chromosomal fusion, fission, and inversions) will be lower than for homozygotes. Information on the degree of adaptation of variants to their environment is included in the objective function. The model simulates populations in two overlapping or adjoining ecological niches, and individuals can move between the two niches. The calculations discussed below consider a hypothetical rapidly reproducing organism with a potential population doubling time of 10 weeks.

Chromosome fusion or fission will result in only 50% of gametes having correct chromosome segregation in meiosis, with the other 50% having potentially lethal abnormalities in gene dosage (Fig. 4). The overall fertility of heterozygotes is thus assumed to be reduced by up to 50%. Offspring homozygous for chromosomal rearrangements are assumed to have unimpaired fertility. Initial simulations treated population growth as exponential, until the carrying capacity of the environment was reached, after which the net growth rate became zero.

## Programming

The speciation algorithm has been implemented in a computer program written in the R language. Source code is included in the supplementary material and can be found here: https://github.com/HiteshBMistry/Speciation. Graphics were generated using R graphics, or the open source program, gnuplot (*Janert, 2010*).

## RESULTS

### Effects of chromosome fusion in a species occupying a single ecological niche

Initially, to explore the population dynamics, we assume a small pre-existing fraction of the karyotypic variants in the population, without at this stage inquiring how they got there. Simulation 1 (Table 1) assumed that birth and death rates for individuals carrying the additional kinetochore were unchanged from those of the original population (shown as Z1 in the tables and figures). However, as discussed in Fig. 4, the efficiency of formation of viable gametes will be reduced by 50% in the heterozygous variant population, Z2. Reproductive efficiency of the homozygous variant population, Z3, is unimpaired. Figure 5 shows that the heterozygous variant population, Z2, increased at half the rate of the original population in the early stage of the growth curve. It was assumed that death rates for all populations remained unchanged until the carrying capacity of the environment (the asymptotic population, AP) was reached, at which point the death rate increased abruptly to equal the birth rate. Alternative growth curves are explored later. Heterozygotes (Z2) as a fraction of the total population declined from the outset. When the carrying capacity of the environment, AP, was reached (at 39 days in Table 1; call this the asymptotic time, AT) the total population levelled off. Z2 then declined in absolute numbers. Homozygous

**Table 1 Evolutionary dynamics of chromosomal variants in a single ecological niche.**

| Simulation no. | Initial populations: | | | Doubling time | | Death rate | | Ta (weeks) | Te (weeks) |
|---|---|---|---|---|---|---|---|---|---|
| | Z1 | Z2 | Z3 | (homozygotes) | (heterozygotes) | (wild-type) | (variant) | | |
| 1 | 10,000 | 100 | 0.2 | 10 | 20 | 0.01 | 0.01 | 39.0 | 220.0 |
| 2 | 100 | 80 | 20 | 10 | 20 | 0.01 | 0.01 | 120.5 | 380.5 |
| 3 | 100 | 0 | 200 | 10 | 20 | 0.01 | 0.01 | 116.0 | 393.0[a] |
| 4 | 100 | 100 | 100 | 10 | 20 | 0.01 | 0.01 | 138.5 | – |
| 5 | 100 | 0 | 100 | 10 | 20 | 0.01 | 0.01 | 148.0 | – |
| 6 | 100 | 0 | 99 | 10 | 20 | 0.01 | 0.01 | 148.0 | 639.5 |
| 7 | 10,000 | 100 | 0.2 | 10 | 20 | 0.05 | 0.05 | 123.5 | 220.0 |
| 8 | 10,000 | 100 | 0.2 | 10 | 10.5 | 0.01 | 0.01 | 39.0 | 2,305.5 |
| 9 | 10,000 | 100 | 0.2 | 10 | 20 | 0.0 | 0.0 | 33.5 | 220.0 |
| 10 | 10,000 | 100 | 0.2 | 10 | 20 | 0.07 | 0.07 | not reached | 48.5 |
| 11 | 10,000 | 1 | 0 | 10 | 20 | 0.05 | 0.01 | 123.5 | 1,790.0[a] |
| 12 | 10,000 | 1 | 0 | 10 | 20 | 0.05 | 0.01 | 123.5 | 43.5 |
| 13 | 10,000 | 1 | 0 | 10 | 20 | 0.05 | 0.017 | 123.5 | 2,738.0 |
| 14 | 10,000 | 1 | 0 | 10 | 20 | 0.05 | 0.015 | 123.5 | 7,360.0[a] |

**Notes.**

Simulations calculated populations of wild-type homozygotes (Z1), heterozygotes (Z2), and variant homozygotes (Z3). Growth was assumed to be exponential until the carrying capacity of the habitat was reached, after which the total population remained constant.

Doubling times in weeks; death rate, fraction of population dying per week. Where the death rate of wild-type and variant homozygotes differs, the death rate for heterozygotes will be that of the chromosomal variants for dominant mutations, and of the wild-type for recessive mutations. Ta, time to carrying capacity; Te, time to extinction of variants (except for simulations 3 and 11, where the wild-type becomes extinct).

[a] Extinction of wild-type.

variants (Z3) declined in absolute numbers from the outset (if, as seems unlikely, Z3 was ever nonzero: nonintegral values of population sizes may be interpreted as probabilities). After AT the rate of decline of Z3 increased. Z2 fell below 0.5 at 220 days. This will be referred to as the extinction time, ET.

Simulation 2 explores the effect of founder population size. The population of Z1 is now two logs smaller than it was in Table 1, and the combined initial population of homozygous and heterozygous variants (Z2 + Z3) is now equal to Z1. However, the variants still became extinct, though it took longer. If the initial population of homozygous chromosomal variants, Z3, was larger than the wild-type population, then the variants became the dominant population, and the wild-type moved towards extinction (simulation 3). Simulation 4 shows the extreme case where the initial numbers of Z1, Z2, and Z3 were identical. In this case the variant did not become extinct, and the ratio of the three karyotypes in the asymptotic population was the Mendelian 1:2:1. This state, in which wild-type and variant maintain equal populations, is unstable: a transient decrease in either of the homozygous populations will result in its eventual extinction. Note that changing the initial number of heterozygotes did not change the outcome, except by increasing the time to the asymptote (simulation 5). To summarize simulations 1–5 of Table 1: when the growth parameters (doubling time and death rate) for wild-type and variant are identical, whichever has the smallest population will eventually become extinct. The reason for this is that the heterozygous population has equal numbers of wild-type and

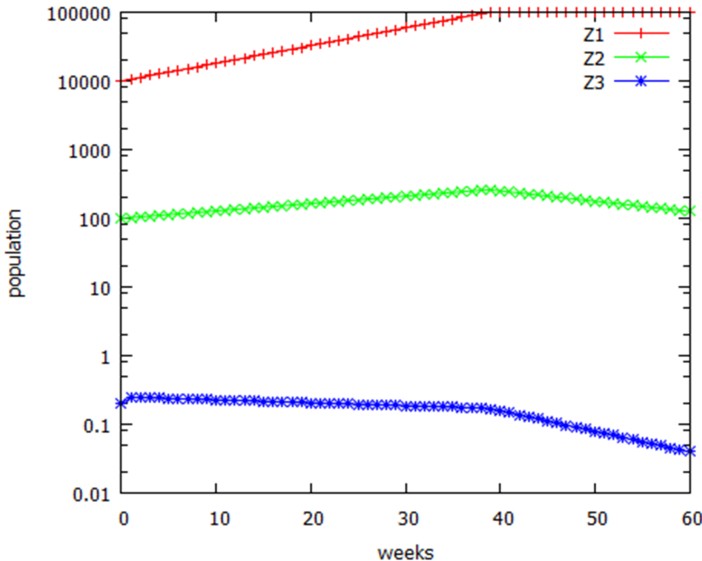

**Figure 5** **Growth kinetics.** Growth of wild-type (Z1), chromosomal variant (Z3), and heterozygous (Z2) populations assuming a low initial population of heterozygotes, and exponential growth that ceases when the total population reaches the carrying capacity of the habitat.

variant chromosomes, and thus a greater *proportion* of whichever population is the smaller. Because the heterozygotes have lower fertility, this results in an ever-decreasing proportion of whichever chromosome type starts out with the smaller population. In a situation where the chromosomal variant results from a spontaneous re-arrangement, the variants will always be outnumbered, so that, other things being equal, variants will always become extinct. This is how the SAC maintains karyotypic stability.

Even a slight population disadvantage of the variants, however, resulted in their extinction (Table 1, simulation 6 and Table S1). Simulation 7 assumed that the death rate for all populations was increased by a factor of 5. This resulted in >3-fold increase in time to asymptote, but time to extinction of the variant population was unchanged. Simulation 8 made the more conservative assumption that the heterozygotes had only a slight fertility disadvantage, with a potential population doubling time of 10.5 weeks, only 5% longer than normal. The variants still became extinct, though it took much longer. Increasing the death rate to 0.07/ week or higher resulted in extinction of the entire population (Table S1). Decreasing all death rates to zero decreased the time to asymptote to 33.5 weeks, but time to extinction of the variants was unaffected (simulation 9). Increasing the death rate to 0.07/wk resulted in extinction of the entire population (simulation 10).

Simulation 11 (Table 1) now assumed that the karyotypic variant carried a dominant mutation that conferred a survival advantage in the original ecological niche. Time to asymptote was slightly decreased, and the variant now dominated the entire population, with the wild-type eventually becoming extinct. If the wild-type population originally existed in two or more geographically separated environments, then the variant, which is now both reproductively and geographically isolated from the wild-type, will have become

**Table 2   Effect of different growth curves on evolutionary dynamics of wild-type and chromosomal variant populations.**

| Growth curve | $T_{0.85}$ | Te (weeks) |
|---|---|---|
| Exponential[a] | 43.0 | 255.0 |
| Gompertzian[b] | 43.0 | 1,951.5 |
| Logistic[c] | 43.0 | 133.5 |

Notes.

Growth of wild-type and chromosomal-variant organisms was modelled in a single niche, and times to extinction of the variant were modelled using different growth equations.

Growth parameters were adjusted to give comparable time to 85% ($T_{0.85}$) of asymptote for all growth curves.

[a] Doubling time (homozygotes) 11.6 weeks, (heterozygotes) 23.2 weeks; death rate (all populations) 0.07/week.

[b] Doubling time (homozygotes) 1.14 weeks, (heterozygotes) 2.28 weeks.

[c] Doubling time (homozygotes) 5.9 weeks, (heterozygotes) 11.8 weeks.

a new species. This would be an example of allopatric speciation. Simulation 12 assumed the same parameter values as simulation 11, except that the mutation in the karyotypic variant strain was now treated as recessive. In this case, the variant became rapidly extinct. Returning to the assumption that the mutation carried by the variant was dominant, the degree of advantage conferred by the mutation was important. In simulation 13, the decrease in death rate caused by the mutation was less than 3-fold, and the variant still became extinct. Decreasing the death rate of the homozygous and heterozygous variants slightly more, to 0.016, was sufficient to tip the balance in favour of the variant strain (simulation 14).

In practice, chromosomal variants must first present as a single heterozygous individual. Simulation 11 models that situation, and shows that, given a survival advantage sufficient to offset their reproductive disadvantage, the progeny of that single individual will eventually become the majority population.

It may be argued that the growth curve used in the previous simulations was overly simplistic. Growth was assumed to be exponential until the carrying capacity of the ecosystem was reached, at which point growth abruptly became zero. A more realistic scenario can be modelled using a form of the Gompertz equation (Table 2, line 2). Population increase was assumed to reflect a balance between birth and death. Birth rates were treated as constant, but death rates were treated as inversely proportional to the log of resource availability, defined as log distance from the carrying capacity, AP. For a population of 1, the death rate is zero; at the carrying capacity the death rate is a steady-state value, $DR_{ss}$, (equal to the birth rate) and at intermediate populations, Z, the death rate $= DR_{ss} * \log Z/\log AP$. The parental population approached its carrying capacity asymptotically, and the homozygous and heterozygous variants slowly became extinct.

Another growth curve widely used in population studies is the discrete logistic model (*May, 1976*). Using this growth equation in the speciation algorithm, as with the Gompertzian growth curve, changed the time taken for the two populations to diverge, but did not alter the qualitative behaviour of the system (Table 2; line 3).The early part of the normal growth curve (Fig. 6) shows the characteristic logistic inflected shape.
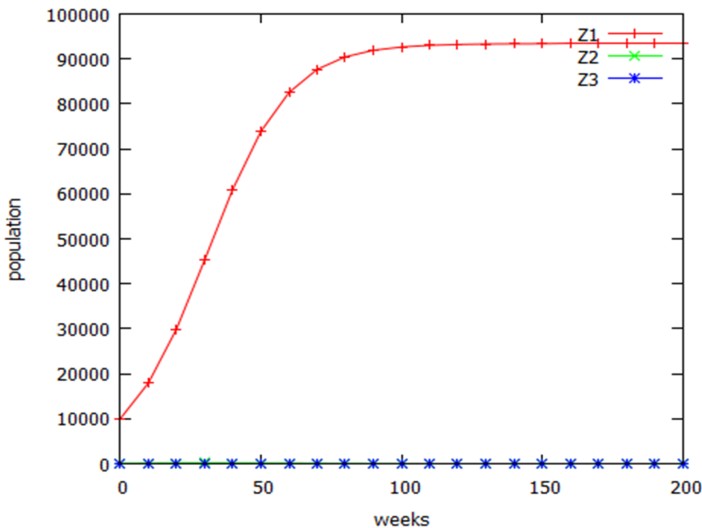

**Figure 6 Logistic growth.** Modelling wild-type (Z1), variant (Z3), and heterozygous (Z2) populations using the discrete logistic growth equation.

## Conclusions from studies with the model of a chromosomal variant in a single ecological niche

Consideration of spindle assembly checkpoint dynamics indicates that a mismatch in chromosome numbers between maternal and paternal karyotypes will result in formation of 50% of normal gametes, and 50% with deleted or duplicated genetic material. These defective gametes, when fertilised, are likely to lead to nonviable zygotes, or zygotes that develop into sterile or partially sterile offspring. Thus, heterozygotes will have fertility that is lower than normal by up to 50%. The degree of reproductive impairment affects the time taken for the variants to become extinct, but does not change the final outcome. This conclusion applies both to chromosomal rearrangement caused by neokinetochore formation (giving an increased chromosome number) and chromosomal fusion (leading to decreased chromosome number). Zygotes that are homozygous for the rearranged chromosomes should segregate normally in mitosis and meiosis, and have unchanged fertility. The qualitative behaviour of the system is independent of the detailed form of the growth curve (exponential, logistic, or Gompertzian). If the variant is assumed to have a significantly lower death rate than wild-type, this may be sufficient to overcome the variant's reproductive disadvantage, and the wild-type may become extinct in its original habitat. For this to happen, the mutation causing the lower death rate must be dominant.

## Simulating the effects of chromosome fusion in a species that can populate two overlapping ecological niches

Subsequent simulations assumed that the organism under study could survive in either of two ecological niches. These niches could be overlapping or adjacent geographically, but differed in some factor that affected survival—temperature, water availability, availability of food or shelter, presence of predators or parasites. The original strain of the organism is

**Table 3  Colonisation of an adjoining niche by organisms originating from a single heterozygote in niche 1.**

| Simulation | Mutation | Migration | Survival factor 1 | Survival factor 2 | Steady-state populations | | | Time to steady state (weeks) |
|---|---|---|---|---|---|---|---|---|
| | | | | | Z4 | Z5 | Z6 | |
| 1 | dominant | yes | 0.01 | 0.01 | 100,030 | 0 | 0 | 160 |
| 2 | dominant | yes | 0.01 | 0.01 | 36 | 1,473 | 98,521 | 660 |
| 3 | dominant | yes (350 weeks) | 0.01 | 0.01 | 0 | 0 | 100,000 | 820 |
| 4 | recessive | yes | 0.01 | 0.010 | 0 | 0 | 0 | 400 |
| 5 | dominant | yes | 0.01 | 1.0 | 36 | 1,473 | 98,521 | 580 |
| 6 | dominant | yes | 0.001 | 0.01 | 3 | 174 | 99,826 | 620 |
| 7 | dominant | yes | 0.01 | 0.01 | 33 | 1,083 | 98,914 | 780 |
| 8 | dominant | yes | 0.001 | 0.01 | 3 | 118 | 99,882 | 980 |

**Notes.**

Population dynamics of wild-type and variant organisms whose range spans covers two adjacent or overlapping niches. Z1, wild-type homozygotes in niche1; Z2, heterozygotes in niche 1, Z3, variant homozygotes in niche 1, Z4, wild-type homozygotes in niche 2; Z5, heterozygotes in niche 2, Z6, variant homozygotes in niche 2.

Doubling times: Z1, Z3, Z4 and Z6: 10 weeks. Z2 and Z5: 20 weeks (except simulations 7 and 8). 40 weeks (simulations 7 and 8).

Death rates: 0.01 for Z1, Z2, Z3 and Z6; Z5, 0.01 except for simulation 4, where it is 0.99; Z4, 0.99, except for simulation 1, where it is 0.01.

poorly adapted to niche 2, and this was reflected in a higher death rate in this environment. The initial assumption was that no organisms lived in niche 2, but that when the carrying capacity of niche 1 was reached, organisms colonised niche 2 (e.g., in search of food). Initially this migration was assumed to be one-way. Table 3, simulation 1 assumed that generation times and death rates did not differ between niche 1 and niche 2. The karyotypic variant became extinct in both niche 1 and niche 2. The total population in niche 2 was slightly greater than the carrying capacity because of immigration.

In Table 3, simulation 1 assumed that the result of modelling the growth of organisms in the two niches, where Z4 (the original karyotype) had a high death rate in niche 2, and where the carrying capacity of niche 2 was limited. Only 1% of migrants survived. Two separate populations emerged, but a stable fraction of heterozygotes remained (Fig. 7). Does this represent speciation? It does in the sense that two stable, karyotypically distinct populations co-existed in the same overall geographical environment. A small population of partially fertile heterozygotes persisted. This is analogous to the situation seen (e.g.,) with big cat species (*Wilson, 2001*).

If we assume that migration into niche 2 ceased after 350 weeks (Table 3, simulation 3), by 600 weeks there were no variant type organisms in niche 1, and no wild-type organisms in niche 2 Depending on our definition, speciation has resulted. The two divergent populations were geographically isolated, so this may be regarded as an instance of allopatric speciation. If we assume that the mutation that accompanied the chromosomal rearrangement was recessive, then speciation did not occur (Table 3, simulation 4).

In Table 3, simulation 5 is a repeat of simulation 2, but now making the assumption that 100% of members of populations 2 and 3 that migrated to niche 2 survived to reproduce. This did not affect the outcome: the karyotypic variant became the dominant population in niche 2, though the steady state population balance was achieved

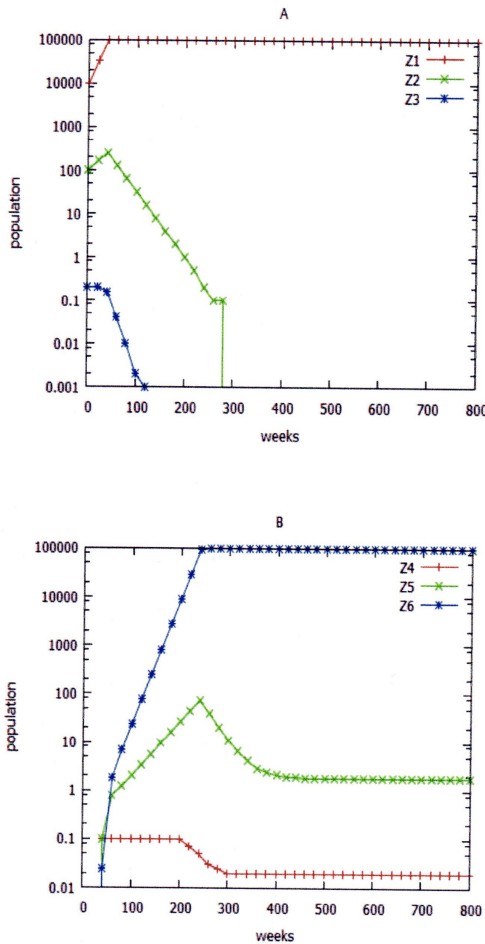

**Figure 7 Peripatric speciation.** Peripatric speciation when the wild-type population is very poorly adapted to the second ecological niche (data of Table 3, simulation 3). Z1, Z2, and Z3 are wild-type, heterozygote, and variant populations, respectively, in niche 1, and Z4, Z5, and Z6 are wild-type, heterozygote, and variant populations, respectively, in niche 2. (A) niche 1; (B) niche 2.

more rapidly than when only a small fraction of the immigrants survived. Changing the survival factor for the wild-type to 0.001, that is, assuming that only one per thousand of migrants into niche 2 survived to reproduce, reduced the number of heterozygotes in niche 2 to <0.2% (Table 3, simulation 6).

If the process of meiosis is inefficient when a pair of short chromosomes has to align with a single long partner, efficiency of the overall process may be decreased below the 50% calculated from the spindle checkpoint kinetics. In Table 3, simulation 7 assumes that efficiency of the pairing process is reduced by 50%, extending the doubling time of heterozygotes (populations 2 and 5) to 40 weeks. Simulation 8 of Table 3 combined inefficient hybridization with low survival factors. This set of parameter values resulted in heterozygotes being decreased to about 0.12% of the total population. According to this model, they can be decreased to an arbitrarily low level but never eliminated.

Simulation 1 in Table 4 modelled the effect of using a different expression for migration rates. Previously the assumption had been made that migration only occurred when the total population in niche 1 was above the carrying capacity:

$$mg[n] = XS * frac[n] * sf[n]$$

where $mg[n]$ is the number of migrants of karyotype n, XS is the excess population above the carrying capacity, $frac[n]$ is the size of population n as a fraction of the total population, and $sf[n]$ is the survival factor for karyotype n, i.e., the fraction of immigrants of karyotype n in niche 2 who survived to reproduce. As an alternative, let migration be inversely proportional to the difference between total population and carrying capacity:

$$mg[n] = Z[n] * Z[0]/AP * sf[n]$$

where $Z[n]$ is the population size of karyotype n in niche 1, $Z[0]$ is the total population in niche 1, AP is the asymptotic population in niche 1 (the carrying capacity), and $mg[n]$ and $sf[n]$ are as defined above. As before, the karyotypic variant became extinct in niche 1, and became the dominant population in niche 2, though the proportion of heterozygotes in niche 2 was larger than when migration was only assumed to occur after the population in niche 1 reached its carrying capacity. This alternative expression for migration rates is used in conjunction with the discrete logistic growth curve. Table S2 shows an example of a single heterozygous cell in niche 1 giving rise to a nascent species in niche 2, even though the variants became extinct in niche 1.

Simulation 2 in Table 4 shows a repeat of simulation 1, but assumes that migration is bi-directional. This model was used to explore 2-way migration, for example with migration in each direction inversely proportional to food availability (i.e., directly proportional to population) or directly proportional to death rate. Survival factors were set to 1.0 for populations 2 and 3, and to 1e−6 for population 1. The variants became the dominant population in niche 2. Subsequent simulations found that the survival factor of the original strain could be as high as $7 \times 10^{-2}$, but values of $8 \times 10^{-2}$ and higher resulted in the variant population becoming extinct in niche 2 (Table S2). In each case the steady state number of heterozygotes in niche 2, sustained by inward migration of wild-type organisms, was very low, but non-zero. Allowing back-migration had the paradoxical effect of increasing the steady state populations of wild-type (Z4) and heterozygotes (Z5) in the population of niche 2. Comparing simulation 2 with simulation 1 (Table 4), this seems to be because the back migration caused a higher steady state population of heterozygotes (Z2) in niche 1 (and to a much lesser extent of the homozygous variants, Z3).

Simulations 4 and 5 of Table 4 compare the migration from niche 1 to niche 2 of chromosomal variants associated with a mutation that confers a survival advantage in niche 2. Even a recessive mutation (simulation 5) was able to establish a population of variant homozygotes in niche 2, though the number of heterozygotes was much lower than for simulation 4, where the mutation was dominant.

In Table 4, simulations 6 and 7 modelled the situation where both niches were partially occupied before the chromosomal rearrangement occurred. The model showed that a low level of occupancy of niche 2 by the wild-type did not alter the final steady-state
**Table 4** Population dynamics in two adjoining ecological niches modelled by a discrete logistic growth curve.

| Simulation | Mutation | Death rates | | | | | | Reverse migration | Survival factor 1 | Survival factor 2 | Steady-state populations | | | | Time to steady state (weeks) |
|---|---|---|---|---|---|---|---|---|---|---|---|---|---|---|---|
| | | Z1 | Z2 | Z3 | Z4 | Z5 | Z6 | | | | Z2 | Z4 | Z5 | Z6 | |
| 1 | dominant | 0.001 | 0.001 | 0.001 | 0.99 | 0.001 | 0.001 | no | 1.0e−6 | 1.0 | 0 | 0.1 | 0.6 | 92,110 | 480 |
| 2 | dominant | 0.001 | 0.001 | 0.001 | 0.99 | 0.001 | 0.001 | yes | 1.0e−6 | 1.0 | 1 | 0.1 | 133 | 92,051 | 420 |
| 3 | dominant | 0.001 | 0.001 | 0.001 | 0.99 | 0.001 | 0.001 | yes | 0 | 1.0 | 4 | 0 | 133 | 92,051 | 460 |
| 4 | dominant | 0.010 | 0.010 | 0.010 | 0.99 | 0.010 | 0.010 | yes | 1.0e−6 | 1.0 | 3 | 0.1 | 87 | 80,231 | 420 |
| 5 | recessive | 0.010 | 0.010 | 0.010 | 0.99 | 0.99 | 0.010 | no | 1.0e−6 | 1.0 | 3 | 0.1 | 2.6 | 80,273 | 450 |
| 6 | dominant | 0.060 | 0.060 | 0.060 | 0.99 | 0.060 | 0.060 | yes | 1.0e−6 | 1.0 | 0.1 | 0 | 0.2 | 10,363 | 2,080 |
| 7 | dominant | 0.070 | 0.070 | 0.070 | 0.99 | 0.070 | 0.060 | yes | 1.0e−6 | 1.0 | 0 | 0 | 0 | 0 | 2,950 |

Notes.

Population dynamics of wild-type and variant organisms whose range spans covers two adjacent or overlapping niches. Z1, wild-type homozygotes in niche1; Z2, heterozygotes in niche 1, Z3, variant homozygotes in niche 1, Z4, wild-type homozygotes in niche 2; Z5, heterozygotes in niche 2, Z6, variant homozygotes in niche 2.

Doubling times: Z1, Z3, Z4 and Z6: 10 weeks. Z2 and Z5: 20 weeks.

Initial population counts: Z1, 10,000; Z2, 1; Z3, 0; Z4, 0 (except for simulations 6 and 7); Z5; 0, Z6, 0. Simulations 6 and 7, Z4 = 1,000.

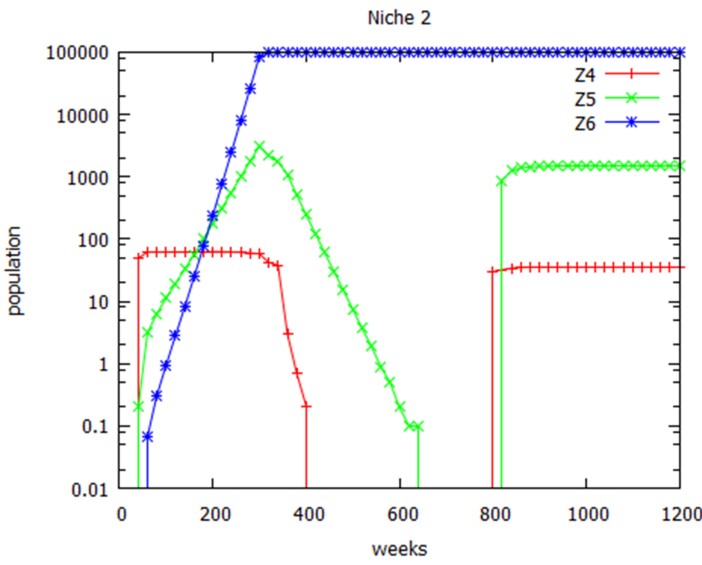

**Figure 8 Geography and colonization.** Geographical isolation followed by re-colonization. Populations of wild-type (Z4), heterozygotes (Z5) and variant homozygotes (Z6) are shown in niche 2. Migration from niche 1 to niche 2 was assumed to occur until week 350, after which it stops, but resumes at 800 weeks.

distribution, although it took longer to reach (simulation 6). The outcome was dependent on the initial conditions: a small increase in the death rate resulted in the variant population slowly becoming extinct (simulation 7).

### Simulating geographical isolation

In Table 3, simulation 3 predicted that if migration from niche 1 to niche 2 was completely stopped after 350 weeks, that only the wild-type strain would survive in niche 1, and only the variant strain would survive in niche 2. Geographical separation of two populations thus resulted in different variants dominating in the two environments, based upon their different adaption to their respective environments. It may be argued that this in itself does not represent speciation, because there is nothing to stop the two populations subsequently interbreeding and forming partially fertile heterozygotes. The new strain in niche 2 has become, in the terminology of *Bush (1994)*, a host race. The present analysis, however, has shown that, in the absence of continued migration from niche 1, the wild-type organisms and heterozygotes in niche 2 will become extinct. Figure 8 shows the result of repeating simulation 3 of Table 3, but now allowing migration from niche 1 to niche 2 to resume, after the original karyotype had become extinct in niche 2. A low level of heterozygotes now re-appeared, and an even lower level of the original homozygous strain. This shows that the continued existence of heterozygotes in niche 2 was dependent upon continued immigration.

## DISCUSSION

Inaccurate chromosome segregation in germ line cells *must* confer a selective disadvantage. If it were otherwise, stable species could not exist. For a karyotypic variant to diverge

from the ancestral line requires *either* a compensating environmental advantage (which could be conferred by absence of the ancestral line, as occurs in allopatric speciation) *or* a genetic advantage. *Dawkins (1987)* and *Dawkins (1995)* presents the current consensus view of speciation as being driven primarily by geographic separation. He argues that "gene flow but not much" is necessary between habitats (*Dawkins, 2004*). Clearly some gene flow is required for additional habitat to be colonized, but if the level is too high, the two populations will be sufficiently mixed that they will not diverge. Simulations in the present study support this conclusion. On p. 320 of *Dawkins (2004)* he makes the statement "it isn't always a geographic separation". If an adjoining or overlapping niche is hostile to the wild-type organism, but less so to the variant, reproductive isolation may occur despite close geographical proximity (pp. 423-424 of *Dawkins (2004)*). Again, the present modelling study has shown that reproductive isolation, even if incomplete, can drive peripatric speciation. Our studies support the conclusion that geographical isolation is not necessary for speciation, though it may be an enabling factor. Reproductive isolation, however, is both necessary and sufficient.

The conclusions that chromosomal rearrangement in meiosis is a primary driver of speciation, and that the dynamics of the SAC provide the main mechanism of those rearrangements, rest upon the following premises:

> Pairs of acrocentric chromosomes resulting from neokinetochore formation are capable of meiotic pairing, with at least partial efficiency, with the cognate mediocentric chromosome from which they were derived. Alternatively, a single chromosome resulting from chromosome fusion (with subsequent loss of one kinetochore) is capable of meiotic pairing, with at least partial efficiency, with the cognate chromosomes from which it was derived.
>
> The SAC in such pairings will delay anaphase until one kinetochore in each chromatid is correctly attached. If, in the chromatid with two kinetochores, one is incorrectly attached, the SAC will still register tension, and such cells may proceed to anaphase.
>
> 50% of gametes resulting from such pairings will have a full genetic complement, and will be fully fertile. 25% of gametes will lack part of the genome, and are likely to be infertile. The remaining 25% of gametes will have part of the genome duplicated, resulting in abnormal gene dosage. This may result in spontaneous abortion or in birth defects. The overall fertility of heterozygotes between wild-type germ cells and those resulting from the two specific kinds of chromosomal rearrangements discussed in (1) will thus be >50%<100%. This reduced fertility will result in the chromosomal rearrangement being eventually eliminated by natural selection, even though the fertility of homozygotes carrying the rearrangement is unimpaired.
>
> If the rearranged chromosome resulting from chromosomal fusion, or one of the acrocentric chromosomes resulting from neokinetochore formation carries a mutation that enhances survival, this survival advantage may be sufficient to counter the decreased fertility of heterozygotes. Once

sufficient numbers of rearranged homozygotes appear, since they do not have decreased fertility, they will be favoured by selection and become the dominant population, while remaining the same species.

If the rearranged chromosome resulting from chromosomal fusion, or one of the acrocentric chromosomes resulting from neokinetochore formation carries a mutation that enhances survival in a different, but accessible, ecological niche, the possibility exists that the variant will become the dominant population in that second niche, while the wild-type remains the dominant population in the original habitat. The two populations, while occupying overlapping or adjacent geographical territories, have different chromosome numbers, are partially reproductively isolated, and may now be considered different species.

It is possible that chromosomal rearrangements other than fission or fusion could result in decreased fertility, and resulting partial reproductive isolation of the variants, and that this could lead to peripatric speciation. However, only fissions and fusions that result in partial reproductive isolation determined by the dynamics of the SAC, as discussed above, will result in pairs of related species whose karyotype differs in number by a single chromosome pair. These qualitative conclusions are independent of the equations used to describe population growth or the migration process.

Modelling such a complex organelle as the SAC involves simplifying assumptions. In fact, both tension and attachment are signalled, but our analysis of the effects of chromosomal fission or fusion has focussed upon tension. Configurations in Figs. 2A and 2B are under tension, and according to our model would proceed to anaphase. However, they contain unattached kinetochores that may continue to produce a wait signal. The effect of this would be to produce fewer gametes of type IV and VI than indicated in Fig. 3, and following fertilization, fewer zygotes of type (3) and (4) as shown in Fig. 4. The model may thus over-estimate the effect of chromosomal fusion and fission on fertility. We have repeated our calculations assuming that the fertility of heterozygotes is decreased by less than 50%. In each case, the qualitative outcome (extinction or speciation) was the same, though the time course was different. For example, if a population of 10,000 individuals with a doubling time of 10 weeks contained a single heterozygote resulting from chromosomal fusion and the fertility deficit was 50%, so the variant doubling time was 20 weeks, the variant became extinct in 20.5 weeks. If we instead assumed only a 20% decrease in fertility of heterozygoes, resulting in doubling time of 12.5 weeks, the variants were extinct at 50.5 weeks.

It may be objected that this mechanism of speciation requires the unlikely simultaneous occurrence of two events: a chromosomal rearrangement, and a germ-line mutation. In fact, chromosomal rearrangements during meiosis are not particularly rare. It is estimated that 20% of all human pregnancies spontaneously abort, and it is likely that a high proportion of these are accounted for by karyotypic abnormalities (*Bradley, Johnson & Pober, 2006*). In any event, speciation events are very rare, and do not appear to be out of line with the

frequency predicted by the present analysis. A possibility not considered by our model is that chromosomal rearrangements could sometimes provide a selective advantage (or disadvantage) without the involvement of a mutation on the rearranged chromosome; this could happen, for example, if a gene were brought by the rearrangement under the control of a different promoter.

The suggestion that chromosomal rearrangement is a primary driver of speciation has a long history (*Ruffié, 1986*). Dobzhansky, in particular, argued that different chromosomal numbers in each parent would interfere with meiosis, causing sterility of the hybrids (*Dobzhansky, 1941*). He argued that chromosomal rearrangement, allelic mutations, environmental changes, and isolation of populations all contributed to speciation. The reaction of Dobzhansky's contemporaries, and the role of mathematical modelling in assessing competing hypotheses of speciation, have been critically reviewed by *Schwartz (1999)*. The principal objection to chromosomal rearrangement as a driver of speciation has been that such rearrangements result in reproductive incompatibility with the ancestral karyotype, so it is difficult to see how they can be propagated. *Parris (2011)* and *Parris (2013)* has argued that the probability of simultaneous identical rearrangements may sometimes be sufficient to allow these "hopeful monsters" to find a mate. The present model, based upon the partial fertility of heterozygotes, largely supports Dobzhansky's main conclusions, but carries the argument forward by showing that a consideration of the dynamics of the SAC is necessary to build a consistent model that incorporates these various contributions to speciation.

Despite persistent suggestions (*Eldredge & Gould, 1972*; *Jones, 1993*; *Dennett, 1995*; *Arnold, 2013*) that Darwin did not explain the origin of species, the present analysis is consistent with natural selection being a primary driving force of speciation. The essence of Darwinism is that competition for resources between related variants is determined by an objective function (natural selection). When the related variants are germ-line mutations or recombinants resulting from crossing-over during meiosis, natural selection results in adaptation of a species to its environment. When the variants are the result of chromosomal rearrangements with resulting partial reproductive isolation, the result is speciation.

The function of the SAC is to ensure descent *without* modification. The mechanism by which it achieves this is complex and error-prone. It deals with errors by making them lethal, and therefore irrelevant. Most of the time—sometimes for millions of generations—this works very well, and ensures the integrity of the species. Most of the time, genetics can focus on the genes present in the genome, without being concerned with how they are packaged. Once in a great while, this quality control process is undermined by a simultaneous, compensating error that allows heterozygotes expressing a small sub-set of chromosomal rearrangements to persist and multiply. The result is two stable, reproductively isolated populations (species) with different karyotypes. The dynamics of the SAC have been selected to maintain stable karyotypes. It is the rare events that subvert this process that explain the origin of species and make it possible for evolution of sexually reproducing species to occur.

Other computational models of speciation have emphasised the importance of gene flow (*Kondrashov & Kondrashov, 1999*; *Savolainen et al., 2006*; *Bolnick, 2006*; *Nosil, 2008*;

*Malinsky et al., 2015*). Our explanation of speciation differs from earlier models in placing the emphasis on reproductive isolation caused by chromosomal changes. We do not discount the role of gene flow: indeed, our model shows that karyotypic changes can only propagate in the presence of genetic mutations that confer a sufficient survival advantage to counter the inherent reproductive disadvantage of the chromosomal rearrangements.

Based on the interpretation of speciation discussed here, the critical difference between humans and the great apes is not brain size, or bipedal gait, or the use of tools. It is the fact that humans have 23 pairs of chromosomes, and the apes, with an almost identical complement of genes, have 24 pairs. The difference resulted from fusion of the ape chromosomes 2a and 2b (*Wienberg et al., 1994*; *Müller & Wienberg, 2001*). The first hominin was the first individual to have that karyotype and pass it on to fertile offspring. This could only have happened if the chromosome fusion was accompanied by a mutation that was able to counter that chromosomal rearrangement's reproductive disadvantage. What was that mutation?

## CONCLUSIONS

Chromosomal rearrangements that are accompanied by mutations may result in peripatric speciation. A consideration of the dynamics of the mitotic spindle checkpoint suggests that although chromosomal fusions or fissions may cause impaired fertility, gametes from rearranged karyotypes can form heterozygotes with gametes from the parental karyptype that contain a normal gene complement. If the rearranged zygotes carry a mutation that confers a selective advantage in an environment adjacent to the original habitat, the heterozygotes may generate a population that is reproductively isolated from the parental line, and eventually give rise to organisms homozygous for the chromosomal rearrangement, which are now a new species.

## ACKNOWLEDGEMENTS

We are grateful to Dr William Bains and Dr Christophe Chassagnole for helpful advice and discussion.

### Funding
The authors received no funding for this work.

### Competing Interests
Robert C. Jackson is employed by Pharmacometrics Ltd. Hitesh B Mistry has no competing interests.

### Author Contributions
- Robert C. Jackson conceived and designed the simulations, performed the simulations, analyzed the data, prepared figures and/or tables, authored or reviewed drafts of the paper, and approved the final draft.

- Hitesh B. Mistry analyzed the data, authored or reviewed drafts of the paper, and approved the final draft.

## Data Availability

Code for the algorithm is available at GitHub: https://github.com/HiteshBMistry/Speciation.

## Supplemental Information

Supplemental information for this article can be found online at http://dx.doi.org/10.7717/peerj.9073#supplemental-information.

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
