# Peer review of "The spindle assembly checkpoint and speciation"

_PeerJ, doi:10.7717/peerj.9073_

## Round 0.1 · original submission · Major Revisions

I found the paper very interesting and stimulating from the point of view of evolution. The hypothesis that sympatric speciation may be driven by reproductive isolation caused by chromosomal rearrangements and differential selection in two partially overlapping niches is attractive but it has been criticized on the ground that heterozygotes arising from two different karyotypes are not viable. The main novely of this work consists in the assumption that the heterozygotes may indeed be viable. The authors nicely show through mathematical modeling that this hypothesis together with the differential selection results in two distinct species adapted to the two different niches. However the two reviewers, who are expert in the spindle point, are not convinced that the viability of the heterozygotes can be predicted from what is currently known about the SAC dynamics, therefore I think that it is better to present this as a plausible assumption and to follow the detailed suggestions of the reviewers when describing the properties of the SAC.

·

Basic reporting

This manuscript explores the idea that chromosomal rearrangements during mitosis may be a primary driver of speciation. This is not a new hypothesis and previous criticisms were mostly based on the reproductive incompatibility with the ancestral karyotype. The authors in this study however claim that considering partial fertility of heterozygous may allow the separation of a new species as they show using mathematical modeling. They also introduce the concept of dynamics of the SAC (whose molecular regulation has been deeply studied in the last decade) as a factor contributing to speciation. Overall, this is an interesting topic that is well introduced and discussed in the manuscript.

Experimental design

Overall, I don’t have specific comments on the mathematical modeling as it is beyond my expertise. My comments mostly related to some definitions used in the introduction and related conclusions.

Validity of the findings

SAC. Lines 75-95. The authors basically present the SAC as a mechanism that senses lack of tension, in part through the activity of aurora kinases. This is not strictly wrong. However, experts in SAC do not have a complete agreement on whether the SAC senses tension or attachment. I would say that most authors prefer the attachment as the major molecular event monitored by the SAC. Accordingly, kinases that monitor attachment such as MPS1 are core components of the SAC. Many authors do not consider aurora kinases as core components of the SAC, but complementary activities that induce de-attachment upon reduced tension. This discussion does not change the main message of the manuscript, but it may be convenient to introduce the discussion between attachment and tension.

Even if the previous discussion is properly discussed in the manuscript, I still have problems understanding the relevance of the SAC in speciation (especially given the relevance given to SAC in the title, abstract, introduction and part of the discussion). In fact, they claim that their study is novel when compared to classic papers because they have introduced SAC dynamic concepts in their work. However, I think the authors could have written the same manuscript and obtaining the same conclusions without referring to the SAC at all. The main assumption for their conclusions is the partial fertility of the heterozygous, but this is equally possible whatever SAC properties are. I cannot evaluate the mathematical expressions in their algorithm but I really wonder whether any specific SAC property has been used there.

In addition, many SAC properties are really different in different species. For instance, SAC ablation in mitotic cells does not induce any deficiency in flies but it is lethal in mammals. Again, I have the impression that this is not really relevant for the manuscript because SAC properties are not really part of the algorithm used.

Some sentences are difficult to understand or are perhaps not properly sustained by the data or concepts used in the manuscript. For instance (line 422) “the dynamics of the SAC provide the main mechanism of those rearrangements”.. I would say that these rearrangements happen even despite the SAC, or because the SAC is not efficient enough, but saying that the SAC provides the mechanism for these re-arrangements seems counterintuitive. Perhaps the authors imply that the fact that the SAC is not efficient enough has been selected to allow evolution? But the way these concepts are used is not clear for this reviewer.

Finally, the SAC has been deeply analyzed in mitosis (and by extension very likely may apply to meiosis II), but how the SAC works in meiosis I (the expected scenario for most of the discussion in the manuscript) is not really understood and perhaps the authors may extend more in detail these implications.

The idea that a parallel mutation occurred together with the arrangements is attractive, but I wondered whether the authors have evaluated that the re-arrangements per se are “the” mutation. For instance, re-arrangements can kill genes or place now enhancers in the vicinity of some genes. In some cases, it has been proposed that reduced number of chromosomes may facilitate prometaphase, so perhaps just reducing or increasing the number of chromosomes per se may have some advantages or disadvantages.

Additional comments

N/A

Reviewer 2 ·

Basic reporting

The authors propose a mechanism by which new species can diverge from the existing population through chromosome rearrangements occurring during meiosis, allowing them to become reproductively isolated. They suggest that the dynamics of the SAC provide the mechanism to allow this speciation to occur.
In the introduction, authors state that the SAC is satisfied only by tension, however this is the subject of debate in the current literature. This conclusion has been mainly drawn from experiments using low dose taxol, a drug that stabilises microtubules and decreases inter-sister kinetochore tension. However, importantly, this drug can also generate single unattached kinetochores which are known to generate a SAC signal to delay the cell (Proudfoot et al, 2019 Cell Reports; Roscioli et al, 2019 BioRXiv, figure 4). Therefore it is difficult to separate tension from attachment in terms of SAC activation. Further experiments have shown that SAC proteins are not lost from kinetochores upon gain of tension using inter-sister kinetochore distance as a readout (Kuhn and Dumont, 2019, Journal of Cell Science). While this is not the focus of this manuscript, authors should be aware that SAC dynamics are not yet fully understood and should be cautious of any oversimplification as it does not represent the true biological mechanism.
Furthermore, authors have grossly oversimplified the mechanisms of SAC signalling in the introduction section ‘Normal operation of the spindle assembly checkpoint’ by stating that the wait signal is mediated by Aurora B. While Aurora B is the most upstream kinase involved in SAC signalling, there are many more events occurring after this. Authors must introduce downstream mechanisms, including the Bub and Mad proteins, and how they are recruited and removed from the kinetochore. They should also introduce the generation of Mitotic Checkpoint Complex (MCC), which constitutes the wait anaphase signal. Authors can refer to Musacchio and Salmon, 2007 (reference 9).
I also note that the authors introduce mitotic checkpoint signalling however the context of this analysis appears to be within meiosis. Authors should comment on the differences between mitosis, meiosis I (co-orientation) and meiosis II (bi-orientation) to make this clearer, and how this affects SAC monitoring. I appreciate authors are focusing on meiosis II for this manuscript, which is more similar to a mitotic division, but introduction of the meiosis I segregation event would help make it clearer to readers how chromosome rearrangement events arise. There is a review written by Hiroyuki Ohkura (Meiosis: an overview of key differences from mitosis 2015, Cold Spring Harb Perspect Biol) that may be of interest.
In the discussion section, authors state that chromosomal rearrangements during meiosis are not rare (line 461), but then use the point of Downs syndrome to provide evidence for this in humans. Are they referring to Down’s syndrome as trisomy 21 or translocation Downs syndrome? This is important as Downs syndrome is a result of aneuploidy leading to the egg containing 3 copies of chromosome 21. This is not a chromosomal rearrangement event but instead a chromosome mis-segregation event leading to aneuploidy. Translocation Downs syndrome, a definite chromosomal rearrangement, is much rarer. We know that meiotic aneuploidy (whole chromosome gain/loss) is high in human eggs, meanwhile meiotic aneuploidy is actually rare in other species. See Hassold and Hunt, 2001 Nature Reviews for a good overview on this. Authors need to be more precise on whether they are discussing chromosomal rearrangements (changes within the chromosomes) or aneuploidy (gain or loss of a whole chromosome), as these have distinct mechanisms of generation.
It is also worth considering that aneuploidy in human eggs is known to increase with increasing maternal age (>35). Is this manuscript considering only human evolution or all species?

Other comments:
• Line 72: Typo ‘chromosomas’
• Lines 84-87. Please provide a reference for the statement that there is a 50:50 chance of attaching to each kinetochore, or if this is an assumption of your model it should be stated as such.

Experimental design

I cannot comment on the mathematical analysis done in this paper as it is not an area that I have experience in.
However, authors describe 3 attachments states; monotelic, syntelic and amphitelic (biorientated). There is also a 4th attachment state – merotelic, where one kinetochore is correctly attached while the other is simultaneously bound to microtubules originating from both spindle poles. This is an important state as it is often ‘invisible’ to SAC signalling and as such can result in whole chromosome mis-segregation events. This should be considered in the text and experimental design.

Validity of the findings

Line 431: Authors state that if one chromatid possesses two kinetochores, then the correct attachment of only one of these is sufficient to satisfy the checkpoint based on tension which will allow the cell to progress to anaphase. As stated in my earlier comments, we cannot assume that the SAC only responds to tension, therefore in my opinion this statement is untrue as the incorrectly attached second kinetochore would still signal to the SAC, and this is sufficient to delay anaphase onset (Rieder et al, 1994, Journal of cell biology). Authors need to discuss the relevance of this.

Additional comments

The manuscript has been prepared with good English. I am very intrigued by the authors hypothesis that chromosomal rearrangement and subsequent satisfaction of the SAC in the following meiotic division could be a driver of speciation. This is a very compelling idea and I think could be an important mechanism. However, I am not confident that the analysis presented in this manuscript provides strong evidence to prove this point, as there have been a lot of assumptions made that are not biologically sound. Authors could improve this by adding in merotelic attachment, as I have noted above. Authors also need to be careful with their conclusions bearing in mind that the SAC is not satisfied by tension alone.
More generally, the figures should be better annotated so that the reader can more easily follow the ideas. Figure legends should also provide more detail so that the figures can stand alone from the text and other figures. The tables are also quite hard to interpret (I am not sure if the formatting has changed between the review PDF to the original manuscript).

---

## Round 0.2 · accepted · Accept

Both reviewers found that the paper has improved and its underlying assumptions are better explained. I find the subject very interesting and important in the evolutionary context.

·

Basic reporting

no comment

Experimental design

no comment

Validity of the findings

no comment

Additional comments

I thank the authors for considering all the points raised during the reviewing process. The manuscript is now improved by discussing more clearly the assumptions and alternative interpretations

Reviewer 2 ·

Basic reporting

The manuscript is now much more coherent. Authors have added some further detail on SAC signalling as requested. They have also made clear that the setting of this study is meiosis II and disclosed the assumptions that this has caused them to make.

Experimental design

I am not experienced in mathematical modelling so cannot comment on this. However, the authors have now made the assumptions in their analysis clear, and discussed the caveats of this in greater depth.

Validity of the findings

No comment